# Constrained Parameter Inference as a Principle for Learning

**Nasir Ahmad**                                             *n.ahmad@donders.ru.nl*
*Department of Artificial Intelligence, Donders Institute, Radboud University*

**Ellen Schrader**                                         *e.schrader@donders.ru.nl*
*Department of Artificial Intelligence, Donders Institute, Radboud University*

**Marcel van Gerven**                                      *m.vangerven@donders.ru.nl*
*Department of Artificial Intelligence, Donders Institute, Radboud University*

**Reviewed on OpenReview:** *https://openreview.net/forum?id=CUDdbTT1QC*

## Abstract

Learning in neural networks is often framed as a problem in which targeted error signals are directly propagated to parameters and used to produce updates that induce more optimal network behaviour. Backpropagation of error (BP) is an example of such an approach and has proven to be a highly successful application of stochastic gradient descent to deep neural networks. We propose constrained parameter inference (COPI) as a new principle for learning. The COPI approach assumes that learning can be set up in a manner where parameters infer their own values based upon observations of their local neuron activities. We find that this estimation of network parameters is possible under the constraints of decorrelated neural inputs and top-down perturbations of neural states for credit assignment. We show that the decorrelation required for COPI allows learning at extremely high learning rates, competitive with that of adaptive optimizers, as used by BP. We further demonstrate that COPI affords a new approach to feature analysis and network compression. Finally, we argue that COPI may shed new light on learning in biological networks given the evidence for decorrelation in the brain.

## 1 Introduction

Learning can be defined as the ability of natural and artificial systems to adapt to changing circumstances based on their experience. In biological and artificial neural networks this requires updating of the parameters that govern the network dynamics (Richards et al., 2019).

A principled way of implementing learning in artificial neural networks is through the backpropagation of error (BP) algorithm (Linnainmaa, 1970; Werbos, 1974). BP is a gradient-based method which uses reverse-mode automatic differentiation to compute the gradients that are needed for individual parameter updating (Baydin et al., 2018). This approach relies on the repeated application of forward and backward passes through the network. In the forward (inference) pass, network activity is propagated forward to compute network outputs. In the backward (learning) pass, the loss gradient associated with the network outputs is propagated in the reverse direction for parameter updating.

While effective, BP makes use of the transmission of gradients using biologically implausible non-local operations, and multiple separated network passes (Grossberg, 1987; Crick, 1989; Lillicrap et al., 2020). Alternative approaches, such as Hebbian learning and subspace methods circumvent this problem yet are restricted to unsupervised learning and do not afford (deep) credit assignment (Brea & Gerstner, 2016; Pehlevan et al., 2015).

Here, we propose constrained parameter inference (COPI) as a new principle for learning. COPI uses information that can be made locally available at the level of individual parameters whose values are being inferred under certain constraints. Note that COPI is distinct from methods that rely on measuring gradients through activity differences (see the NGRAD hypothesis (Lillicrap et al., 2020)), in that no difference in activity needs to be computed to determine parameter updates. Specifically, by constructing a mixed network activity state – in the BP case a simple summation of the forward and backward passes for output units – parameters can infer their own optimal values by observation of node activities alone. This is distinct to many proposed biologically plausible methods which require parameters to measure differences in some activity, either physically using separate compartments/signals, or across time between two phases (Bengio, 2014; Scellier & Bengio, 2017; Ernoult et al., 2020; Whittington & Bogacz, 2017; Sacramento et al., 2018; Payeur et al., 2021). Thus, COPI provides a framework which might in future enable online continuous learning where parameter updates are based upon single state measurements.

Furthermore, the COPI algorithm is not tied to any particular credit assignment method. In this sense it assumes that credit can be assigned to units (by a user's method of choice) and simply describes how parameters should update their values given a network state observation. Credit assignment is integrated into COPI by top-down perturbations. The form of the perturbation is precisely what determines which credit-assignment algorithm is being used for learning within the system, whether based on backpropagation, feedback alignment (Lillicrap et al., 2016; Nøkland, 2016), target propagation (Bengio, 2014; Ahmad et al., 2020) or otherwise. Thus, COPI does not address credit assignment as such but rather proposes a general approach for learning based upon a single mixed-state regime.

In the following, we demonstrate that COPI provides a powerful framework for learning which is at least as effective as backpropagation of error while having the potential to rely on local operations only. Moreover, as will be shown, COPI allows for efficient linear approximations that facilitate feature visualisation and network compression. Hence, it also provides benefits in terms of interpretable and efficient deep learning. This has direct implications for modern machine learning as COPI can be used as a replacement for the parameter-updating step in backpropagation applications across a wide range of settings.

## 2 Methods

In this section, we develop the constrained parameter inference (COPI) approach and describe its use for parameter estimation in feedforward neural networks.

### 2.1 Deep neural networks

Let us consider deep neural networks consisting of $L$ layers. The input-output transformation in layer $l$ is given by

$$y_l = f(a_l) = f(W_l x_l)$$

with output $y_l$, activation function $f$, activation $a_l = W_l x_l$, input $x_l$ and weight matrix $W_l \in \mathbb{R}^{K_l \times K_{l-1}}$, where $K_l$ indicates the number of units in layer $l$. As usual, the input to a layer $l > 1$ is given by the output of the previous layer, that is, $x_l = y_{l-1}$. Learning in deep neural networks amounts to determining for each layer in the network a weight update $\Delta_{W_l}$ such that the update rule

$$W_l \leftarrow W_l + \eta \Delta_{W_l}$$

converges towards those weights that minimize a loss $\ell$ for some dataset $\mathcal{D}$ given a suitable learning rate $\eta > 0$. Locally, the optimum by gradient descent (GD) is to take a step in the direction of the negative expected gradient of the loss. That is,

$$\Delta_{W_l}^{\text{gd}} = -\mathbb{E}\left[\nabla_{W_l}\ell\right],$$

where, in practice, the expectation is taken under the empirical distribution.

### 2.2 Constrained parameter inference in feedforward systems

Here, we develop an alternative approach and relate it directly to both stochastic gradient descent and local parameter inference. Note that the key transformation in a deep feedforward neural network is carried out

by a weight matrix given by

$$a_l = W_l x_l \, .$$

Suppose we know the target activation $z_l$ for this transformation. This can be expressed as an alternative transformation

$$z_l = W_l^* x_l$$

for some desired weight matrix $W_l^*$. Ideally, we would like to use a learning algorithm which guarantees convergence of the current weight matrix to the desired weight matrix. A straightforward proposal is to carry out a decay from the current weight values to the desired weight values, such that the weight update is of the form

$$\Delta_{W_l} = \mathbb{E}\left[W_l^* - W_l\right] = W_l^* - W_l \, . \tag{1}$$

Of course, the key goal is to achieve this weight update without making use of the (unknown) desired weights. How to achieve this, is described in the following sections.

### 2.3 Learning the forward weights

Let us rewrite the desired weight matrix in the following way:

$$\begin{aligned}
W_l^* &= W_l^* \left( \mathbb{E}\left[x_l x_l^\top\right] \mathbb{E}\left[x_l x_l^\top\right]^{-1} \right) \\
&= \mathbb{E}\left[W_l^* x_l x_l^\top\right] \mathbb{E}\left[x_l x_l^\top\right]^{-1} \\
&= \mathbb{E}\left[z_l x_l^\top\right] \mathbb{E}\left[x_l x_l^\top\right]^{-1}
\end{aligned}$$

with $\mathbb{E}\left[x_l x_l^\top\right]$ the (sample) autocorrelation matrix. We here assume this matrix to be invertible, though this condition is later shown to be unnecessary. If we plug this back into Eq. (1) then we obtain

$$\Delta_{W_l} = \mathbb{E}\left[z_l x_l^\top\right] \mathbb{E}[x_l x_l^\top]^{-1} - W_l \, , \tag{2}$$

allowing the weight update to be expressed in terms of target outputs, $z_l$, rather than (unknown) desired weights. This is an expression of the least-squares optimization algorithm.

Let us assume for the moment that the inputs $x_l$ are distributed such that they have zero covariance and unit variance, i.e., the inputs are whitened. This implies that the autocorrelation matrix is given by the identity matrix, that is, $\mathbb{E}\left[x_l x_l^\top\right] = I$. In this case, Eq. (2) reduces to the simple update rule

$$\Delta_{W_l} = \mathbb{E}\left[z_l x_l^\top\right] - W_l \, .$$

In practice, it may be unreasonable (and perhaps even undesirable) to assume perfectly whitened input data. A more realistic and achievable scenario is one in which we make the less restrictive assumption that the data is decorrelated rather than whitened. This implies that the autocorrelation matrix is diagonal, and that $\mathbb{E}\left[x_l x_l^\top\right] = \mathrm{diag}\left(\mathbb{E}\left[x_l^2\right]\right)$ with $x_l^2$ the vector of squared elements of $x_l$. Right-multiplying both sides of Eq. (2) by this expression, and assuming that $\mathbb{E}\left[x_l x_l^\top\right]$ is indeed diagonal, we obtain

$$\Delta_{W_l} \mathrm{diag}\left(\mathbb{E}\left[x_l^2\right]\right) = \mathbb{E}\left[z_l x_l^\top\right] - W_l \,\mathrm{diag}\left(\mathbb{E}\left[x_l^2\right]\right) \, .$$

This matrix multiplication amounts to a rescaling of the columns of $\Delta_{W_l}$, and thereby a relative scaling of their learning rates. This finally leads to our constrained parameter inference (COPI) learning rule

$$\Delta_{W_l}^{\mathrm{copi}} = \mathbb{E}\left[z_l x_l^\top\right] - W_l \,\mathrm{diag}\left(\mathbb{E}\left[x_l^2\right]\right) \, , \tag{3}$$

which is solely composed of a Hebbian correlational learning term and weight decay term. COPI receives its name from the fact that there are two constraints in play. First, the availability of a target or 'desired' state $z_l$ for each layer and, second, the requirement that the inputs $x_l$ are decorrelated.

### 2.4 Input decorrelation

We did not yet address how to ensure that the inputs to each layer are decorrelated. To this end, we introduce a new decorrelation method which transforms the potentially correlation-rich outputs $y_{l-1}$ of a layer into decorrelated inputs $x_l$ to the following layer using the transformation

$$x_l = R_l y_{l-1},$$

where $R_l$ is a decorrelating 'lateral' weight matrix.

We set out to reduce the correlation in the output data $x_l$ by measurement of its correlation and inducing a shift toward lower correlation. In particular, consider a desired change in a given sample of the form

$$x_l \leftarrow x_l - \eta \left( \mathbb{E} \left[ x_l x_l^\top \right] - \mathrm{diag} \left( \mathbb{E} \left[ x_l^2 \right] \right) \right) x_l,$$

where the expectations could be taken over the empirical distribution. We can shift this decorrelating transform from the output activities $x_l$ to the decorrelating matrix $R_l$. To do so, consider substituting $x_l = R_l y_{l-1}$, such that we may write

$$
\begin{aligned}
x_l &\leftarrow x_l - \eta \left( \mathbb{E} \left[ x_l x_l^\top \right] - \mathrm{diag} \left( \mathbb{E} \left[ x_l^2 \right] \right) \right) x_l \\
&\leftarrow R_l y_{l-1} - \eta \left( \mathbb{E} \left[ x_l x_l^\top \right] - \mathrm{diag} \left( \mathbb{E} \left[ x_l^2 \right] \right) \right) R_l y_{l-1} \\
&\leftarrow \left( R_l - \eta \left( \mathbb{E} \left[ x_l x_l^\top \right] - \mathrm{diag} \left( \mathbb{E} \left[ x_l^2 \right] \right) \right) R_l \right) y_{l-1}.
\end{aligned}
$$

We converge to the optimal decorrelating matrix using an update $R_l \leftarrow R_l + \eta \Delta_{R_l}^{\mathrm{copi}}$, where

$$
\begin{aligned}
\Delta_{R_l}^{\mathrm{copi}} &= - \left( \mathbb{E} \left[ x_l x_l^\top \right] - \mathrm{diag} \left( \mathbb{E} \left[ x_l^2 \right] \right) \right) R_l \\
&= - \left( \mathbb{E} \left[ x_l q_l^\top \right] - \mathrm{diag} \left( \mathbb{E} \left[ x_l^2 \right] \right) R_l \right)
\end{aligned}
\tag{4}
$$

with $q_l = R_l x_l$. Note that this decorrelating transform can also be derived rigorously as a gradient descent method, see Appendix A.

The update can be made more local and therefore more biologically plausible by an approximation, which we explore here. To make the information locally available for the update of the decorrelation, we assume that this correlation reduction can be carried out in either direction – with the target of correlation reduction and source being exchangeable. This assumption allows us to arrive at a more biologically plausible learning rule given by

$$
\Delta_{R_l}^{\mathrm{bio\text{-}copi}} = - \left( \mathbb{E} \left[ q_l x_l^\top \right] - R_l \, \mathrm{diag} \left( \mathbb{E} \left[ x_l^2 \right] \right) \right)
\tag{5}
$$

with $q_l = R_l x_l$. Equation 5 has exactly the same form as our previous COPI decorrelation rule (3) for learning the forward weights though now acting to update its weights in the same manner as the COPI forward learning rule - using target states $q_l$. These target states are now the total amount of decorrelating signal being provided to a given unit. In effect, the lateral weights are also constantly inferring correlational structure within the activities of a layer of units but, given the negatively-signed update, they update their values to reduce correlation instead.

This rule is more biologically plausible than the theoretically derived decorrelation rule since an update of a lateral weight $r_{ij}$ connecting unit $j$ to unit $i$ relies on $q_i x_j$ rather than $x_i q_j$. That is, an update local to unit $i$ only requires access to its own target state rather than the target state of other units. Furthermore, the weight decay term is now scaled based upon the pre-synaptic activity, which is propagated via the synaptic connection, rather than the post-synaptic activity. As this information is available to the post-synaptic unit and synaptic connection, respectively, it can be used for parameter updating. Both the theoretically derived rule and its more biologically plausible formulation consistently reduce correlation in the output states $x_l$, as shown in Appendix A.

### 2.5 Error signals

Equation (3) expresses learning of forward weights in terms of target states $z_l$. However, without access to targets for each layer of a deep neural network model, one may wonder how this learning approach could be

applied in the first place. To this end, we assume that the target states can be expressed as

$$z_l = a_l + \alpha \delta_l$$

with $\delta_l$ an error signal which perturbs the neuron's state in a desired direction and $\alpha$ a gain term controlling the strength of the perturbation. Note that if we can directly access the $\delta_l$, then the optimal weight change to push future responses in the desired direction $\delta_l$ is simply $\delta_l x^\top$. However, here we assume that the weight-modification process cannot directly access $\delta_l$, but only 'sees' the perturbed activities $z_l = a_l + \alpha \delta_l$. In this case, as explained above, COPI is necessary to produce correct weight changes and push future responses towards these target activities.

Different error signals can be used to induce effective perturbations. According to stochastic gradient descent (SGD), the optimal perturbation is given by

$$\delta_l^{\mathrm{sgd}} = -\frac{d\ell}{da_l}$$

as this guarantees that the neuronal states are driven in the direction of locally-decreasing loss. The error signal at the output layer is given by $\delta_L^{\mathrm{sgd}} = -\frac{\partial \ell}{\partial a_L} = -\mathrm{diag}(f'(a_L))\frac{\partial \ell}{\partial y_L}$. Starting from $\delta_L^{\mathrm{sgd}}$, the layer-specific perturbation can be computed via backpropagation by propagating the error signal from output to input according to $\delta_l^{\mathrm{sgd}} = \frac{\partial a_{l+1}}{\partial a_l}\delta_{l+1}^{\mathrm{sgd}}$ with $\frac{\partial a_{l+1}}{\partial a_l} = \mathrm{diag}(f'(a_l))R_{l+1}^\top W_{l+1}^\top$.

While gradient-based error signals provide a gold standard for the optimal perturbation, we may want to replace this by more biologically-plausible credit assignment methods. These methods typically use the same error signal $\delta_L \triangleq \delta_L^{\mathrm{sgd}}$ for the output layer but propagate the error signal in the input direction using different proposal mechanisms. As an illustration of such an alternative error signal, we consider feedback alignment (FA) (Lillicrap et al., 2016), which supposes that the perturbation from the previous layer can be propagated through fixed random top-down weights $B_{l+1}$ instead of the transposed weights $(W_{l+1}R_{l+1})^\top$, as a way to address the so-called weight transport problem (Grossberg, 1987). Hence, in FA, the layer-wise perturbations are propagated by

$$\delta_l^{\mathrm{fa}} = \mathrm{diag}(f'(a_l))B_{l+1}\delta_{l+1}^{\mathrm{fa}}.$$

In our analyses, we will restrict ourselves to comparing error signals provided by backpropagation and feedback alignment only. Note, however, that other credit assignment methods such as direct feedback alignment (Nøkland, 2016) or target propagation and its variations (Bengio, 2014; Dalm et al., 2023) can be seamlessly integrated in our setup if desired.

## 2.6 Stochastic COPI

Stochastic COPI replaces the expectations over the empirical distributions in Eqs. (3) and (4) by single data points, analogous to stochastic gradient descent (SGD). COPI training on single data points proceeds by computing the stochastic weight updates. For all COPI implementations, the forward weight updates are given by $\Delta_{W_l}^{\mathrm{copi}} = z_l x_l^\top - W_l \,\mathrm{diag}\left(x_l^2\right)$ with target states $z_l = a_l + \alpha \delta_l$, given some suitable error signal $\delta_l$. The decorrelating lateral weight updates are given by $\Delta_{R_l}^{\mathrm{copi}} = -\left(x_l q_l^\top - \mathrm{diag}\left(x_l^2\right)R_l\right)$ with $q_l = R_l x_l$. In practice, as usual, we train on minibatches instead of individual data points. See Algorithm 1 for a pseudo-algorithm which uses gradient-based error signals and a quadratic loss.

For comparison against SGD, it is instructive to consider (stochastic) COPI applied to a single-layer neural network. Recall that the SGD update of a single-layer network is given by $\Delta_W^{\mathrm{sgd}} = -\frac{d\ell}{da}x^\top$. We can manipulate this expression in order to relate SGD to COPI as follows:

$$\begin{aligned}
\Delta_W^{\mathrm{sgd}} &= -\frac{d\ell}{da}x^\top \\
&= \left(a - \frac{d\ell}{da}\right)x^\top - ax^\top \\
&= \left(a + \delta^{\mathrm{sgd}}\right)x^\top - W\left(xx^\top\right).
\end{aligned} \tag{6}$$

---

**Algorithm 1** Constrained Parameter Inference

1: **procedure** COPI(*network*, *data*)
    ▷ *network* consists of randomly initialized forward weight matrices $W_l$ and lateral weight matrices $R_l$ with $1 \leq l \leq L$
    ▷ *data* consists of $N$ input-output pairs $(y_0, y^*)$
    ▷ Parameters: learning rates $\eta_R$ and $\eta_W$; gain term $\alpha$; number of epochs; batch size
2:    **for each** *epoch* **do**
3:        **for each** *batch* $= \{(y_0, y^*)\} \subset data$ **do**
4:            **for** layer $l$ **from** $1$ **to** $L$ **do**       ▷ Forward pass
5:                $x_l = R_l y_{l-1}$       ▷ Decorrelate the input data
6:                $a_l = W_l x_l$       ▷ Compute activation
7:                $y_l = f(a_l)$       ▷ Compute output
8:            **end for**
9:            $\ell = ||y_L - y^*||^2$       ▷ Compute loss
10:            **for** layer $l$ **from** $L$ **to** $1$ **do**       ▷ Backward pass
11:                $\delta_l = -\frac{d\ell}{da_L}$ if $l = L$ else $\delta_l = \frac{\partial a_{l+1}}{\partial a_l}\delta_{l+1}$       ▷ Compute learning signal
12:            **end for**
13:            **for** layer $l$ **from** $L$ **to** $1$ **do**       ▷ Update parameters
14:                $W_l \leftarrow W_l + \eta_W \left( (a_l + \alpha\delta_l)x_l^\top - W_l \operatorname{diag}\left(x_l^2\right) \right)$       ▷ Update forward weights
15:                $R_l \leftarrow R_l - \eta_R \left( x_l(R_l x_l)^\top - \operatorname{diag}\left(x_l^2\right) R_l \right)$       ▷ Update lateral weights
16:            **end for**
17:        **end for**
18:    **end for**
19: **return** *network*
20: **end procedure**

---

This update looks similar to the (stochastic) COPI update $\Delta_W^{\mathrm{copi}} = (a + \alpha\delta^{\mathrm{sgd}})x^\top - W\operatorname{diag}\left(x^2\right)$. The key difference, however, is that, in contrast to SGD, COPI ensures that the inputs are decorrelated, as realized by the COPI update $\Delta_R^{\mathrm{copi}}$. Therefore, the weight decay term is unaffected by sample-wise input cross-correlations. The weight decay term for COPI is Hebbian in nature since the update of $w_{ij}$ relies on $[W\operatorname{diag}\left(x^2\right)]_{ij} = w_{ij}x_i x_j$, which is local to the synapse. In contrast, SGD is non-Hebbian in nature since it relies on $[W(xx^\top)]_{ij} = \sum_k w_{ik}x_k x_j$, which is not local to the synapse.

## 2.7 Linear approximation of non-linear transformations

An additional benefit of the decorrelating properties of COPI is that it allows for the efficient approximation of one or more non-linear transformations by a linear matrix. As will be shown later, this has applications in interpretable and efficient deep learning.

Let us consider a neural network as a non-linear transformation $y = f(x)$. We are interested in computing its linear approximation, given by $y = Bx$. Suppose we have access to empirical data $X = \left[x^{(1)}, \ldots, x^{(N)}\right]$ and $Y = \left[y^{(1)}, \ldots, y^{(N)}\right]$ such that $y^{(n)} = f\left(x^{(n)}\right)$ is the $n$-th input-output pair. The standard ordinary least squares solution for $B$ is given by

$$B = YX^\top \left(XX^\top\right)^{-1} .$$

However, this requires the computation of a matrix inverse, which can be prohibitive for large matrices.

Networks trained using COPI can instead make use of the property that inputs are decorrelated by construction. This implies that $\left(XX^\top\right)^{-1} = \operatorname{diag}\left(1/x_1 x_1^\top, \ldots, 1/x_M x_M^\top\right) \triangleq C$ with $x_m$ the $m$th row of $X$. This allows us to compute a transformation from such decorrelated inputs as

$$B = YX^\top C .$$

Hence, in networks with decorrelated layer-wise inputs or activities, such as produced by COPI, linear approximations of transformations can be efficiently computed without computing inverses. We will use this property in Section 3.2 for both feature visualization and network compression.

Note that this is ultimately the mechanism by which COPI also operates, though in a sample/batch-wise manner with a changing output distribution due to fluctuating learning signals. Specifically, consider the COPI algorithm at its fixed point, such that $\Delta_{W_l}^{\text{copi}} = \mathbb{E}\left[z_l x_l^\top\right] - W_l \operatorname{diag}\left(\mathbb{E}\left[x_l^2\right]\right) = 0$. Under this condition, we can re-arrange to obtain

$$W_l = \mathbb{E}\left[z_l x_l^\top\right] \operatorname{diag}\left(\mathbb{E}\left[x_l^2\right]\right)^{-1},$$

which is equivalent to our above formulation though under the assumption of a fixed desired output $z_l$.

## 3 Results

In the following, we analyse both the convergence properties of COPI and the benefits of the decorrelated representations at every network layer.

### 3.1 COPI performance on standard benchmarks

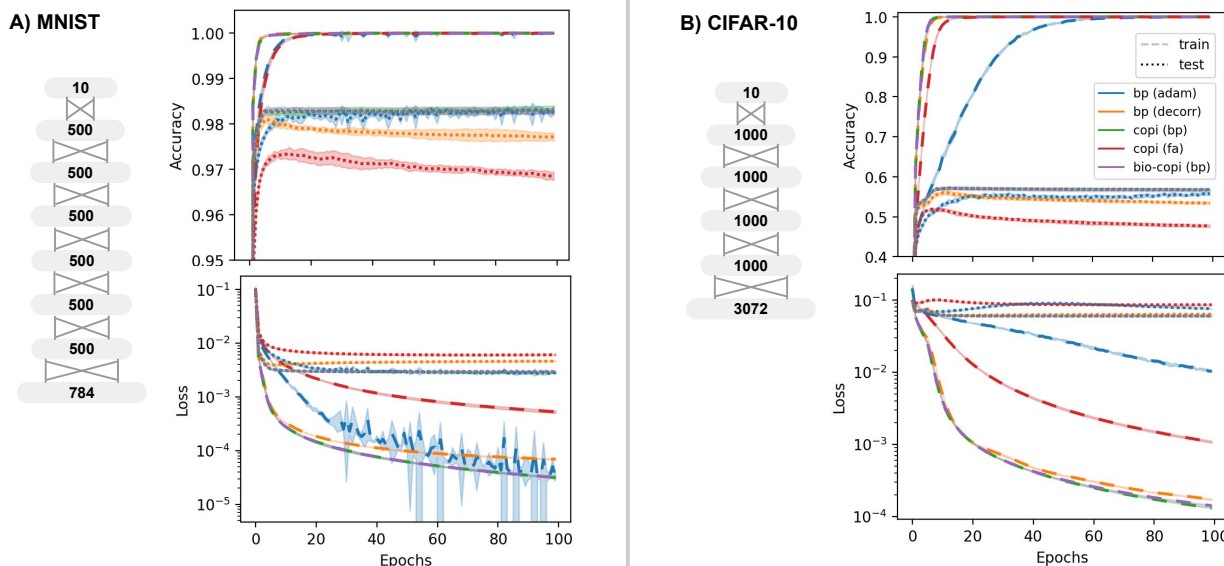

Figure 1: **COPI vs BP performance on standard computer vision classification tasks**. A) Train/test accuracy and loss of a seven-layer (see graphical depiction), fully-connected, feedforward deep neural network model trained and tested on the handwritten MNIST dataset. B) Train/test accuracy of a five-layer, fully-connected, feedforward deep neural network model trained and tested on the CIFAR-10 dataset. All networks were run with five random seeds and envelopes show standard deviation across these networks.

To validate COPI as a principle for learning, we compared it against backpropagation by training fully-connected deep feedforward neural networks trained on the MNIST handwritten digit dataset (LeCun et al., 2010) and the CIFAR-10 image dataset (Krizhevsky & Hinton, 2009).

COPI was simulated together with both gradient-based ($\delta_l^{\text{sgd}}$) and feedback alignment-based ($\delta_l^{\text{fa}}$) perturbations with a loss function composed of the quadratic loss between network outputs and the one-hot encoded labels as 'desired' output. To clarify the benefits of the COPI decorrelation process, we additionally trained networks in which the forward weights $W_l$ were updated by BP using vanilla SGD and lateral weights $R_l$ were introduced and trained by the COPI decorrelating algorithm, labelled 'BP (decorr)'. Furthermore, we obtained baselines with backpropagation alone (no decorrelation) combined with the Adam optimizer (Kingma & Ba, 2014). Figure 1 shows the results of these simulations using learning parameters as described in Appendix B.

Figure 1 shows that gradient-based COPI learning is extremely effective. During training, COPI achieves higher accuracy and lower loss than even an adaptive optimization approach (Adam) on the more challenging CIFAR-10 dataset. When BP is combined with decorrelation, training loss remains consistently lower for COPI across both datasets. We can only attribute this benefit to the explicit difference in the forward COPI and BP rules, where COPI relies on the built-in assumption that the inputs have a decorrelated form. During testing, we observe that COPI slightly outperforms BP (adam) in terms of accuracy on CIFAR-10 and performs consistently better than BP with decorrelation on both datasets. The performance of the more biologically plausible BIO-COPI variant is close to identical to that of regular COPI. COPI learning using feedback alignment was also feasible albeit less effective, consistent with literature (Nøkland, 2016). This demonstrates that different error signals can easily be plugged into the COPI framework when desired. Note further that results generalize to different network depths, as shown in Appendix C, as well as to other loss functions, as shown in Appendix D for the cross-entropy loss.

Table 1: **Peak performance (accuracy) measures of COPI vs BP for the results presented in Figure 1**. Also provided in brackets is the mean epoch at which the networks reached 99% peak performance.

| Method | Peak Performance $\pm$ Standard Dev. (Mean # Epochs to 99% of Peak) | | | |
| | MNIST | | CIFAR-10 | |
| | train | test | train | test |
| --- | --- | --- | --- | --- |
| bp (adam) | $1.0 \pm 0.0\,(6)$ | $\mathbf{0.9838 \pm 0.0004\,(5)}$ | $0.9998 \pm 0.0001\,(53)$ | $0.5619 \pm 0.0023\,(36)$ |
| bp (decorr) | $1.0 \pm 0.0\,(3)$ | $0.9812 \pm 0.0009\,(3)$ | $1.0 \pm 0.0\,(8)$ | $0.5616 \pm 0.0047\,(8)$ |
| copi (bp) | $1.0 \pm 0.0\,(3)$ | $0.9834 \pm 0.0007\,(3)$ | $1.0 \pm 0.0\,(7)$ | $0.5729 \pm 0.0016\,(10)$ |
| copi (fa) | $1.0 \pm 0.0\,(7)$ | $0.9740 \pm 0.0010\,(4)$ | $1.0 \pm 0.0\,(13)$ | $0.5207 \pm 0.0022\,(6)$ |
| bio-copi (bp) | $1.0 \pm 0.0\,(3)$ | $0.9835 \pm 0.0009\,(3)$ | $1.0 \pm 0.0\,(8)$ | $\mathbf{0.5730 \pm 0.0040\,(9)}$ |

Arguably, the largest gain is obtained in terms of convergence speed when using COPI as a learning mechanism. In general, we find that (BIO-)COPI and decorrelating BP (which uses the COPI mechanism for decorrelation) are able to learn much faster than conventional BP with an adaptive optimizer (Adam). As can be seen in Table 1, models employing decorrelation reach close to peak performance (within 99% of peak performance) much more rapidly. This is not due to the used learning rate since performance drops at higher learning rates when using Adam.

## 3.2 Decorrelation for feature analysis and network compression

The COPI algorithm's requirement for decorrelation at every network layer is not only a restriction but also proves beneficial in a number of ways. We explore the decorrelation, as well as the analyses and computations that it enables.

The proposed decorrelation method produces a representation similar to existing whitening methods, such as ZCA (Bell & Sejnowski, 1997). Figure 2A provides a visualisation of a randomly selected set of data samples from the MNIST dataset. From top to bottom are shown: the unprocessed samples, samples processed by the first decorrelating layer of a network trained on MNIST with the COPI algorithm (see MNIST networks described in Figure 1A), and finally a visualisation of samples transformed by a ZCA transform computed on the whole training set. As can be seen, there is qualitative similarity between the COPI- and ZCA-processed data samples. Remaining differences are attributed to the fact that COPI does not scale the individual elements of these samples (pixels) for unit variance, i.e. whitening, but instead produces decorrelation alone in a distributed fashion.

Beyond the input transformation, COPI allows for visualisation of features deeper in the network by exploiting decorrelated layer-wise inputs. That is, we may use the decorrelated input training dataset $X$ and corresponding unit activations $A_l$ to form a linear approximation $A_l = B_l X$ of the receptive field of units deep in the network using the procedure described in Section 2.7. Here, the $i$-th row of $B_l$ provides an average, linear, feature response for unit $i$ in layer $l$ of the network.

Figure 2C shows such extracted features from a random selection of 100 units from the second, fourth and sixth layer and 10 units from the seventh layer of a network trained using COPI on MNIST. These results

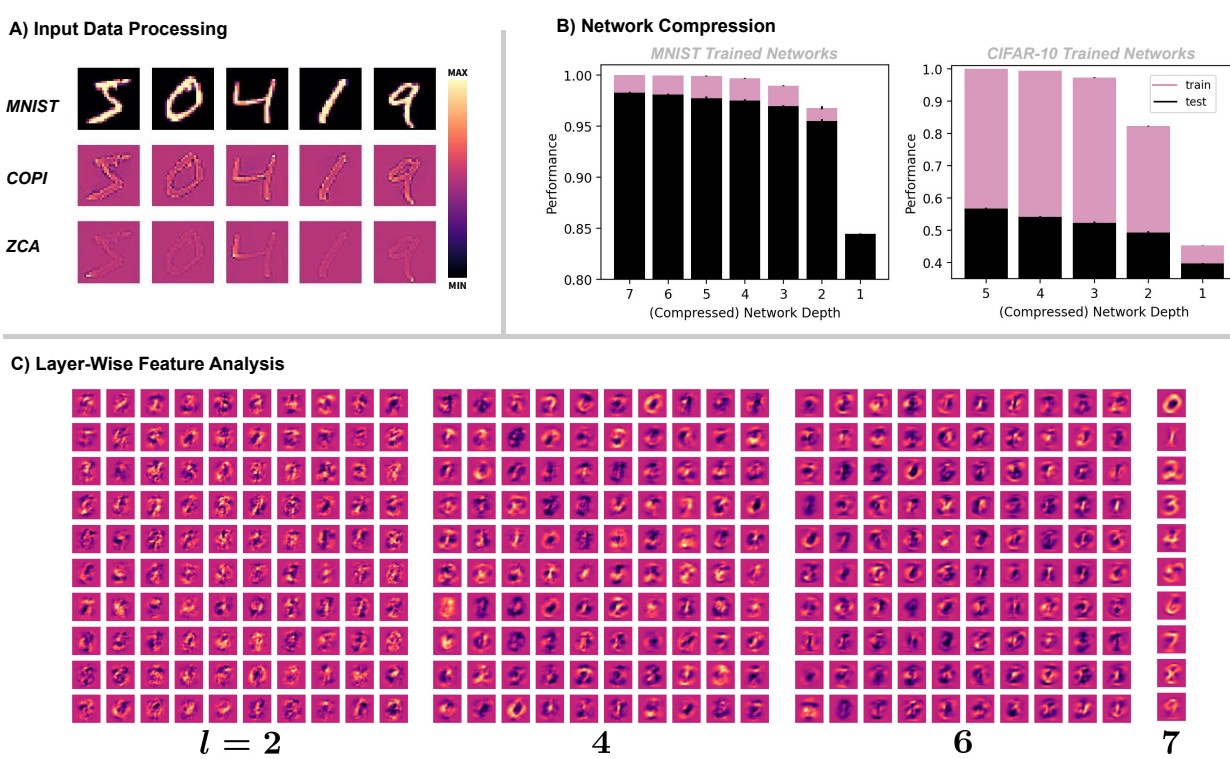

Figure 2: **The effect and utility of decorrelation within COPI network layers for feature readout and network compression**. A) Visualisation of the MNIST dataset (top), the decorrelating transformation produced by the COPI algorithm (middle), and, the whitening transformation produced by ZCA whitening (bottom). B) The decorrelated layer-wise inputs of COPI-trained networks could also be used to efficiently infer linear mappings between distant layers of the network. This allows the removal of network layers and replacement with an inferred, linear approximation of those intermediate layers. Plotted are the performances of the seven-layer MNIST trained COPI networks (left) and five-layer CIFAR-10 trained COPI Networks (right) from Figure 1. This network compression process is repeated for five, randomly seeded networks and error bars show standard deviation across these repeats. Layers are removed from the output layer backwards. Here, the left-most bars (seven/five layers for MNIST/CIFAR-10) correspond to the initial unmodified networks, which are provided for comparison. C) Using the decorrelated network inputs of the COPI networks (see middle row in A), the decorrelated inputs, $x$, from the entire training set data could be (pixel-wise) correlated with the network's layer-wise outputs, $a_l$ (node-wise). This produced a linear approximation of the units' preferred features from different network layers.

are from a single network used to produce the results shown in Figure 2A. This allows us to observe an increasingly digit-oriented feature preference for units as we go deeper into the network, as well as the presence of digit/anti-digit-selective units.

This same mechanism of producing a linear approximation of receptive fields also provides a computationally efficient method to approximate the transformation produced by multiple layers of a COPI-trained network with a linear matrix. That is, we may employ the COPI principle to infer a linear matrix $B_l$ approximating the transformation across multiple network layers such that $A_l \approx B_l X_k$ where $X_k$ is the decorrelated output of some layer $k < l$. This approach allowed us to rapidly convert MNIST and CIFAR-10 trained networks into networks consisting of any smaller number of layers, effectively providing a straightforward approach for network compression.

Figure 2B shows the performance impact of such conversions for the seven-layer network trained on MNIST depicted in Figure 1A (MNIST trained) and the five-layer network trained on CIFAR-10 shown in Figure 1B.

Note that this approximation is done in a single step using the network's response to the training data and does not require any retraining. Network performance is shown to stay relatively high despite the approximation and removal of layers. In fact, for CIFAR-10, we even see that this approximation returns some small gain in test-set performance. Note that layers are removed sequentially from the end of the network and, as can be seen, there is a significant drop in performance when the all layer of each network are approximated, indicating that the transformation in the first layer is crucial for achieving high performance levels. This is consistent with the change in performance when retraining networks consisting of a smaller number of layers, as shown in Appendix C.

## 4 Discussion

In this paper, we introduced constrained parameter inference as a new approach to learning in feedforward neural networks. We derived an effective local learning rule and showed that, under the right conditions, individual weights can infer their own values. The locality required the removal of confounding influences between unit activities within every layer of the neural network, and to this end, we derived an efficient decorrelation rule. We further assumed error signals were available to perturb unit activations towards more desirable states from which the system could learn.

The resulting algorithm allowed us to effectively train deep feedforward neural networks, where performance is competitive with that of backpropagation for both gradient-based and feedback alignment signals. Furthermore, our setup enables much higher effective learning rates than are possible than with vanilla BP and thus allows us to learn at speeds exceeding those possible even using adaptive optimizers. This may contribute to reducing carbon footprint when training large network models (Strubell et al., 2019). The algorithm also allows for more interpretable deep learning via the visualisation of deep decorrelated features (Rudin, 2019; Ras et al., 2022) and could contribute to efficient deep learning as it facilitates network compression (Wang, 2021). Going forward, it is important to expand the tasks to which COPI is applied and investigate its application to a broader class of network architectures. For example, COPI is in principle compatible with other network components such as convolutional layers, but requires careful consideration as for how to carry out decorrelation in an optimal manner.

From a theoretical standpoint, COPI relates to unsupervised methods for subspace learning (Oja, 1982; Földiák & Young, 1998; Pehlevan et al., 2015). In particular, the form of the learning rule we propose bears a resemblance to Oja's rule (Oja, 1982), though it focuses on inference of parameters in the face of perturbations instead of latent factor extraction. See Appendix E for a comparison.

Aside from unsupervised methods, the inference of parameters based upon input and output activities has been previously proposed to overcome the weight-transport problem (Akrout et al., 2019; Ahmad et al., 2021; Guerguiev et al., 2019). In particular, these methods attempt to learn the feedback connectivity required for backpropagation via random stimulation of units and a process of weight inference. Our method similarly attempts to carry out weight inference, but does so without random stimulation and with the purpose of learning of the forward model through combination with top-down perturbations.

It is also interesting to note that our decorrelating mechanism captures some of the key elements of batch normalization (Ioffe & Szegedy, 2015; Huang et al., 2018). First, vanilla batch-normalization makes use of demeaning, a natural outcome of our decorrelation. Furthermore, whitening of batches has been recently shown to be an extremely effective batch-wise processing stage, yielding state-of-the-art performance on a number of challenging classification tasks (Huang et al., 2018), and reduction of covariance between hidden unit activities has been found to be a generalisation encouraging regularizer (Cogswell et al., 2015). However, unlike all of these methods, our method is not batch-computed and is instead a fixed component of the network architecture, learned over the course of the whole dataset and integrated as a network component.

COPI may also shed light on learning and information processing in biological systems. There is both experimental and theoretical evidence that input decorrelation is a feature of neural processing through a number of mechanisms including inhibition, tuning curves, attention, and eye movements (Franke et al., 2017; Bell & Sejnowski, 1997; Pitkow & Meister, 2012; Segal et al., 2015; Vogels et al., 2011; Abbasi-Asl et al., 2016; Cohen & Maunsell, 2009; Dodds et al., 2019; Graham et al., 2006). In particular, center-surround

filters of the LGN appear to produce a form of whitening. Whitening also appears to be key for sparse coding of visual inputs (King et al., 2013). To what extent there is decorrelation between all units projecting to a neuron is of course questionable, though COPI has the potential for modification to account for global or local correlations. The more biologically plausible decorrelation rule described in Section 2.4 suggests how the decorrelation rule here might operate in a local fashion.

Beyond this, inhibitory and excitatory balance (Denève & Machens, 2016) has been formulated in a fashion which can be viewed as encouraging decorrelation. Learning rules which capture excitatory/inhibitory balance, such as the one by Vogels et al. (2011), rely on correlative inhibition between units, which in turn reduce the inter-unit covariance. Such detailed balance has been observed across cortical areas and so it does not seem unreasonable to consider this as a method to encourage decorrelation of not just the input but also downstream 'hidden' layers of neural circuits.

When considering biological plausibility, the current work assumes that error signals are available and do not interfere with ongoing network activity. This means that we rely on a two-phase credit assignment process. For a fully online implementation, the error machinery should be integrated into a single mixed pass, which is an area for future exploration.

We conclude that constrained parameter inference allows for efficient and effective training of deep feedforward neural networks while also providing a promising route towards biologically plausible deep learning.

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

## A COPI decorrelation as gradient descent

In the main text, we provided a description of the custom learning rule for decorrelation which forms a part of the COPI learning approach. Here we expand upon this description and frame the same derivation in terms of gradient descent upon a specific loss function.

The COPI learning algorithms require a decorrelated input, meaning that our decorrelation method should minimise the off-diagonal values of $\mathbb{E}[xx^T]$, where $x$ represents the (vector) input data to any given layer and the expectation is taken empirically over a whole dataset. To this end, we can define an element-wise quadratic loss function $l_i$, representing the total undesirable correlation induced by a unit, indexed $i$, with respect to all other units, indexed $j$, within a single sample such that:

$$l_i = \frac{1}{2} \sum_{j:\ j \neq i} (x_i x_j)^2 \ .$$

The derivative of this expression can then be taken with respect to unit $i$, in order to identify how to modify the activity of unit $x_i$ in order to reduce this loss. Specifically,

$$\frac{\partial l_i}{\partial x_i} = \sum_{j:\ j \neq i} (x_i x_j)\, x_j \,,$$

showing that, via stochastic gradient descent, we can produce greater decorrelation by computing the product between unit activities and removing a unit-activity proportional measure from each unit, $x_i \leftarrow x_i - \eta \frac{\partial l_i}{\partial x_i}$, where $\eta$ would be a learning rate. Vectorizing this stochastic gradient descent across all units allows us to write an update for our data $x$ such that

$$x \leftarrow x - \eta \left( xx^T - \operatorname{diag}\left(x^2\right) \right) x \,,$$

with learning rate $\eta$ and $\operatorname{diag}(\cdot)$ as used in the main text, indicates constructing a square matrix of zeros with the given values upon the diagonal. Finally, as in the main text, we can assume that $x$ is constructed from some transformation, $x = Ry$, such that we can recast this update in terms of the decorrelating transformation matrix, $R$, where

$$Ry \leftarrow Ry - \eta \left( xx^T - \operatorname{diag}(x^2) \right) Ry \quad \Rightarrow \quad Ry \leftarrow \left[ R - \eta \left( xx^T - \operatorname{diag}(x^2) \right) R \right] y \,,$$

providing an equivalent to our derived update rule for decorrelation $\Delta_R^{\text{copi}} = -\eta \left( xx^T - \operatorname{diag}(x^2) \right) R$, as introduced in the main text.

One may ask why we constructed the specific decorrelation rule described above, rather than using an alternative existing rule. For that matter, one may ask why we chose to take the derivative of our decorrelation loss with respect to unit activities, $x$, when deriving this rule instead of directly with respect to the parameters of the transformation matrix, $R$. First, the used derivation allowed the production of a simple, Hebbian-like update and allowed us to formulate, admittedly by approximation, similar and elegant learning rules for forward and lateral weight matrices. This was important as a promising start in order to work toward methods for online and local learning of these transformations. Second, on a more rigorous note, the learning rule we propose produces reductions in inter-unit correlations which are not affected by the scale (eigenvalues) of the matrix $R$. This is a property that is induced by our choice of taking the derivative of our decorrelation loss with respect to the unit activities, $x$, rather than the matrix elements of $R$. Note that we can take the derivative of our above loss with respect to a single element of our decorrelating matrix $R_{ij}$ in the following manner,

$$\frac{\partial l_i}{\partial R_{ij}} = \frac{\partial l_i}{\partial x_i} \frac{dx_i}{dR_{ij}} = \sum_{j:\ j \neq i} (x_i x_j)\, x_j y_j \,.$$

However, reducing correlations by taking the full derivative with respect to the elements of $R$, or via alternative existing methods which have been proposed for decorrelation through simple anti-Hebbian learning (Földiák, 1990; Pehlevan et al., 2015), result in a reduction in correlation which is affected by the scale of the matrix $R$.

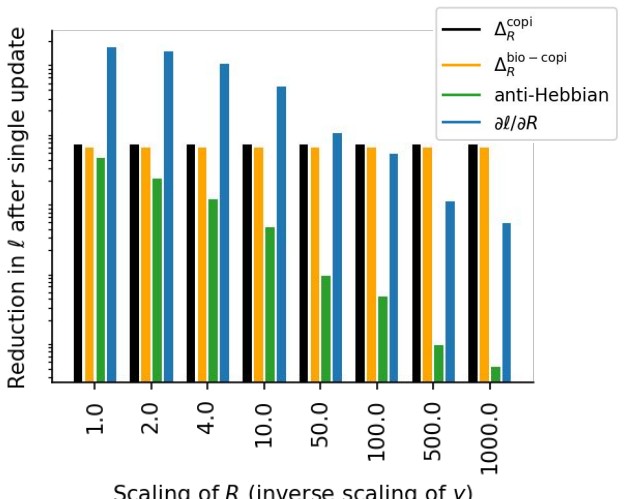

Figure 3: **The reduction in correlation in a a toy-dataset when leveraging various decorrelation rules.** To produce this plot, a dataset was randomly sampled from a multi-variant gaussian distribution and an initial decorrelating matrix $R$ also sampled. This data, with samples $y$, is processed by the decorrelating matrix $R$ to form outputs, $x$. A set of methods were then used to compute a single update to the decorrelating matrix, $R$, and the magnitude of reduction in the loss function (mean of loss $\ell = ||xx^{\top} - \mathrm{diag}(x^2)||_2^2$ over all datapoints in the dataset) was computed and plotted here. Various scalings were then applied to the matrix $R$ and input data $y$, which maintained the output data such that $x = (cR)(y/c)$, and the process of computing the efficacy of different learning rules repeated. As shown, only the proposed COPI decorrelating method produces a consistent reduction in the loss function, regardless of the scale of matrix $R$ or the input data $y$. It is for this reason that this rule is desirable when applied in conjunction with other learning rules which require a relative scaling. The y-scale of this plot is omitted since its scale is arbitrarily dependent upon the initial sampling of data, and this plot is intended to be illustrative.

We can show this effect empirically in set of simple simulations measuring the magnitude of correlation reduction induced by various learning rules, see Figure 3. In order to produce these results, we first construct a dataset by randomly sampling from a 100-dimensional multivariate normal distribution with a randomly sampled (arbitrary) covariance matrix. We further initialised a matrix $R \in \mathbb{R}^{100 \times 100}$ composed of the identity function, $I$, plus random noise added to every element drawn from a [-0.1,0.1] uniform distribution. This matrix, $R$, is used to process the input data, $y$, in order to attempt to produce a decorrelated output $x = Ry$ as in the methods of the main part of this paper. In order to simulate an alternative scaling of the matrix $R$ without affecting the output data distribution, we simulate a rescaling of $R$ by removing a constant factor from the input data and scaling $R$ by this factor, $x = (cR)(y/c)$ With this setup, we could then demonstrate how various methods for learning the matrix $R$ (with various scalings applied) reduce the loss function,

$$\mathcal{L} = \frac{1}{N} \sum_{n=1}^{N} \ell^{(n)} = \frac{1}{N} \sum_{n=1}^{N} \left( x^{(n)} \left( x^{(n)} \right)^{\top} - \mathrm{diag} \left( \left( x^{(n)} \right)^2 \right) \right)^2 ,$$

where $n$ indexes the $N$ samples in the empirical dataset. In Figure 3, the COPI learning rule for decorrelation is compared to the derivative of this loss with respect to the elements of matrix $R$, $\partial l_i / \partial R_{ij}$ above, and also against a simple anti-Hebbian learning rule (Földiák, 1990; Pehlevan et al., 2015), where $\Delta_R^{\text{anti-hebbian}} = -(xx^T - \mathrm{diag}(x^2))$.

As can be seen, the propose COPI learning rule is the only decorrelating learning rule which reduces the loss function by a consistent amount given some output distribution for $x$, regardless of the relative scaling of the decorrelating matrix $R$ and the input data $y$. Having such a decorrelation method, free from a learning rate interference through the scale of matrix $R$ or the unused pre-decorrelation variable $y$, is crucial for the COPI learning system since the forward learning rule and decorrelating learning rules interact and must be

balanced in their speed of learning to avoid runaway correlations affecting the efficacy of the forward learning rule.

## B   Learning setup and parameters

Note that for all simulations which made use of decorrelation (all COPI networks and BP with decorrelation), the networks were first trained for a single epoch with only the decorrelation rule active. This allowed the network to reach a decorrelated state (the desired state for this learning) before forward weight updating was started. All hidden layers used the leaky rectified linear unit (leaky-ReLU) activation function. Training was carried out in mini-batches of size 50 for all simulations (stochastic updates computed within these mini-batches are averaged during application). Network execution and training is described in Algorithm 1. Learning rate parameters are described in Table 2. All code used to produce the results in this paper is available at: https://github.com/nasiryahm/ConstrainedParameterInference.

Table 2: **Parameters for CIFAR-10 and MNIST trained networks (cf. Figure 1).**

| Parameter | BP (Adam) | COPI (with BP/FA gradients) / BP (with decorr) |
|---|---|---|
| LeakyReLU negative slope | 0.1 | 0.1 |
| Learning rate $\eta_W$ | 0.0001 | 0.0001 |
| Learning rate $\eta_R$ | 0.0001 | 0.0001 |
| Gain parameter $\alpha$ | 1.0 | 1000.0 |
| Adam parameter $\beta_1$ | 0.9 | - |
| Adam parameter $\beta_2$ | 0.999 | - |
| Adam parameter $\epsilon$ | $10^{-8}$ | - |

## C   Training networks of various depths

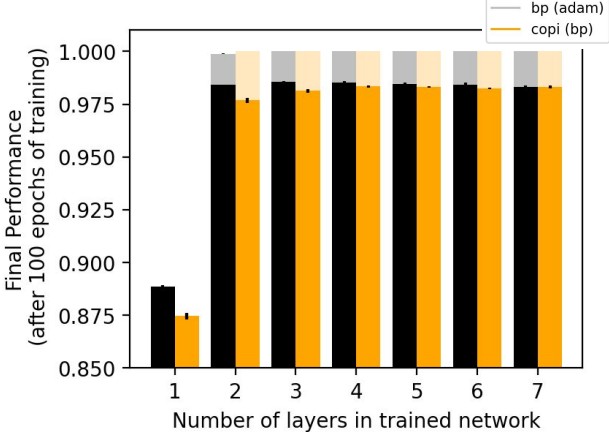

Figure 4: **Train and test accuracy of networks of various depths trained by COPI (BP) vs BP with the Adam optimiser.** Performance of networks ranging from one to seven layers trained and tested using the MNIST handwritten digit set. The networks are composed such that each hidden layer (where present) is composed of 500 units, input layer of size 784, and output layer of size 10.

The main text explored networks with fixed depths. Here we present additional results in which we trained and tested networks between one and seven layers on the MNIST handwritten digit set. These networks were each trained for 100 epochs and their performance measured after training. Parameters used for training followed the setup described in Appendix B.

As can be observed, the networks of extremely shallow depth (one and two layer), have a measurably lower final performance compared to training by BP with the Adam optimizer. We found in our experimentation that this reflects the relative difficulty of decorrelating the input layer of our network, and the importance of the features in the first layer of the network. A similar effect of the importance of the first layer of the network for performance, can be observed in the main text in Figure 2, where, when approximating a deep network with a linear layer, we observed a significant drop in performance when all layers (including the first layer) were removed.

## D    COPI with categorical cross-entropy loss

In the main text, we explored a quadratic loss in the mathematical derivation and simulation. However, this does not imply that COPI can only be applied in the application of the quadratic loss function, in fact any arbitrary loss function applied to the outputs can be used to produce a gradient-based target.

In particular, take any arbitrary loss which is solely a function of the outputs of a network, $\ell = f(y_L)$. By default, we would compute the derivative with respect to the outputs of the network as $\frac{d\ell}{dy_L}$. This arbitrary formulation differs from a quadratic loss with a target, $t_L$, since a quadratic loss is proportional to $y_L - t_L$. However, it is possible to reformulate an arbitrary loss function computed on the outputs in terms of a target in the following manner

$$\frac{d\ell}{dy_L} = \frac{d\ell}{dy_L} + (y_L - y_L)$$
$$= y_L - \left(y_L - \frac{d\ell}{dy_L}\right)$$
$$= y_L - t_L^*$$

where $t_L^*$ is a target formulated for this layer.

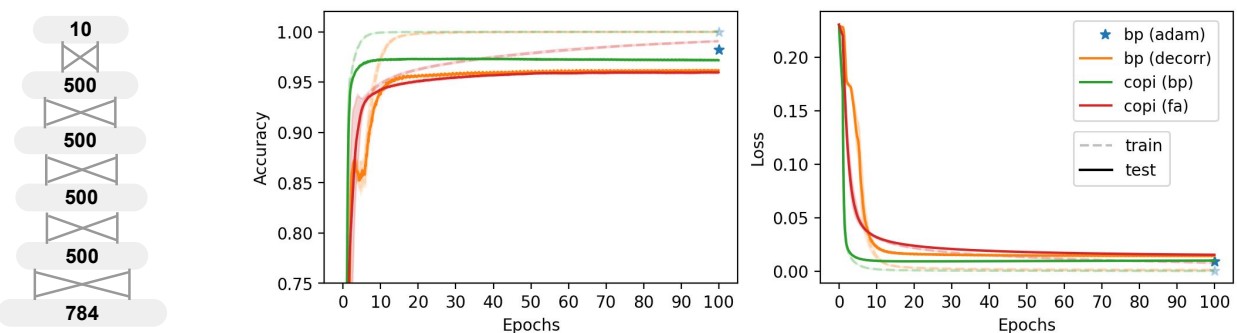

Figure 5: **The performance, measured by training and test accuracy/loss, of a five-layer fully-connected feedforward neural network architecture trained using a categorical cross-entropy loss**. Plotted are various learning approaches combining decorrelation, our algorithm (COPI), and standard stochastic gradient descent by backpropagation of error (BP). All results are shown for training on the MNIST handwritten digit classification task. All networks were run with five random seeds and envelopes show standard deviation across these networks. Results for BIO-COPI not shown given its near identical performance to regular COPI.

We made use of such a target-formulation approach (though in terms of the derivative of the output hidden state $a_L$) to train the same network architecture used to train the networks of Figure 1A with a categorical cross-entropy loss. This again demonstrates a favorable performance and convergence speed when training networks using COPI, though these simulations have not been as thoroughly optimized by parameter search.

## E    Relation between COPI and Oja's rule

Let us consider the (BIO-)COPI update for single synaptic weights, given by

$$\Delta_{w_{ij}}^{\text{copi}} = z_i x_j - w_{ij} x_j^2 = \left( \sum_j w_{ij} x_j \right) x_j - w_{ij} x_j^2 + \delta_i x_j$$

$$\Delta_{r_{ij}}^{\text{copi}} = - \left( q_i x_j - r_{ij} x_j^2 \right) = \left( \sum_j (-r_{ij}) x_j \right) x_j - (-r_{ij}) x_j^2 \, .$$

The first term in both expressions is a Hebbian update which relies on the states of the pre-synaptic units $x_j$ and post-synaptic units $z_i$ or $q_i$ only. The second term in both expressions takes the form of a weight decay. This functional form is similar to Oja's rule (Oja, 1982), which states:

$$\Delta_{m_{ij}}^{\text{oja}} = y_i x_j - m_{ij} y_i^2 = \left( \sum_j m_{ij} x_j \right) x_j - m_{ij} y_j^2$$

for $y = Mx$. COPI differs from Oja's rule in that the forward weight update has an additional term $\delta_i x_j$ and the weight decay for both the forward and lateral weights depends on the (squared) pre-synaptic rather than post-synaptic firing rates. The functional impact of the difference in the weight decay scaling (by post- vs pre-synaptic firing rates), is that Oja's rule scales weight decay in order to normalize the scale of the weights which target a post-synaptic neuron. By comparison, COPI scales the weight decay such as to best infer the scale which the weights should take in order to reproduce the post-synaptic neuron activity given the observed pre-synaptic neuron activity.

