# OpenReview forum: "Constrained Parameter Inference as a Principle for Learning"
_TMLR — Accepted by TMLR_

### Review · Reviewer_UPjV · 2022-12-04

**Summary Of Contributions:**

The paper describes a learning rule in a variant of a classical fully connected multi-layer perceptron. The key difference in the network architecture is to add a matrix $R$ prior to each matrix multiplication with the weights $W$ leading to an effective network activity defined by: $f(WRx)$ where $x$ in the input for this layer. Then the learning rule is defined so that $Rx$ generates a decorrelated (i.e. whitened) version of $x$ while $W$ minimizes the loss function (this defines the BP with decorrelation algorithm reported in the results). To further exhibit a variant of the weights updates for $R$ and $W$ which avoid is "more biologically plausible", the COPI learning rule replaces the covariance $\mathbb{E}[x x^T]$ with its diagonal $diag(x^2)$. The learning rule are then compared on MNIST and CIFAR with many fully connected layers, a strong finding is that decorrelated BP and COPI learn much faster than regular BP for the same performance level.

**Audience:**

Yes

**Claims And Evidence:**

Yes

**Requested Changes:**


For the technical description in the main text: the equations and the text is understandable and clear although I believe it could be written in a lighter fashion. More problematic, I find that the general story of the paper is sometimes detached from the mathematical content and could be improved in three ways:

(1) I find the general framing of the paper around "constrained parameter optimization" to be quite miss leading because I thought it would perform some kind of constrained optimization but the technical part of the paper is not described in this way.

(2) I also think that the paper would from a clearer definition of the meaning of "biologically plausible" learning rule which is solved in this paper with a clear statement about why is that problematic for biology: concretely which terms in BP or BP with decorrelation are problematic and how does COPI solves it? (BIOCOPI is also defined in the appendix, but what is the difference? and why does that matter?)

(3) Unfortunately I did not understand the whole discussion about network compression. I believe that this relates usually to weights pruning or something like that but I did not see how is that relevant for the theoretical method and the experiments. Also I did not understand the figure in the main text about "removed layers". What does that mean in the experiments and why is that interesting?


For the figures reporting the accuracy I would recommend to zoom in in the relevant range (for MNIST 85% - 100%) because this is where everything is happening and I would have liked to see more clearly those learning curves.


Also I cannot see the filters in Figure 2B because there are too small.

**Strengths And Weaknesses:**

I think that the topic of efficient bio-plausible rules for decorrelation in deep networks is important and I find the model $y=f(WRx)$ to be a simple and elegant solution. The plausible update of R which avoids some problem is therefore interesting as well, although I would have liked to know what plausibility it resolves specifically.

I find the relevant on the learning rate of the decorrelated BP and COPI to be very promising and I would love to see a bit more analysis on that. Here are suggestion for control experiments:

(a) what if the learning rate parameter $\eta$ is optimized individually for classical BP and decorrelated BP, would a higher learning rate be more beneficial for BP?  (just reporting the next power of ten to prove it is not that easy to make faster would be sufficient)

(b) I understand that the authors report the result of classical BP as the case with $y=f(WRx)$ with $R$ optimized with SGD. What if $f(Wx)$, is the learning rate better? is the final accuracy on CIFAR better?

---

> ### Author Response · Authors · 2023-01-10
> **Summary of revision by authors.**
>
> Thank you reviewer for you input. Below are our changes and comments in response to your input and requests.
>
> Concerning point (a), we already performed this analysis. We now added the following to the results section:
>
> [...] This is not due to the used learning rate since performance drops at higher learning rates when using Adam.
>
> Specifically, final training loss for BP Adam on CIFAR-10 with a LR of 0.001 is 0.01251 after 100 epochs. For BP Adam with an LR of 0.0001 is 0.01038 (this is a significant difference).
>
> Concerning point (b), BP Adam was trained in the regular way. I.e. using f(Wx). Training using f(RWx) would be slower and have no effect since this is a sequence of linear transformations. Furthermore, the decorrelation matrix, R, is never trained using any supervised data and therefore any benefit of this matrix is aside from the supervised training aspect.
>
> Concerning point (1), please note that the method is called Constrained Parameter Inference as it refers to an inference process acting on the parameters under the constraints of decorrelated input and top-down error signals. Constrained Parameter Optimization would indeed be at risk of being confused with constrained optimization as used in the optimization literature.
>
> Concerning point (2), we moved the section on bio-copi to the main text and describe in more detail why it is more biologically plausible. Particularly, we added:
>
> This rule is more biologically plausible than the theoretically derived decorrelation rule since an update of a lateral weight $r_{ij}$ connecting unit $j$ to unit $i$ relies on $q_i x_j$ rather than $x_i q_j$. That is, an update local to unit $i$ only requires access to its own target state rather than the target state of other units. As this information is available to the post-synaptic unit it can be used for parameter updating.
>
> Regarding our definition of biological plausibility, note that we introduce this in the Introduction when stating: BP makes use of the transmission of gradients using biologically implausible non-local operations, and multiple separated network passes.
>
> Concerning point (3), also see the response to Reviewer ZrgT. We moved relevant material from the appendix to a new Section 2.7 and rewrote it. We also updated the results section. The key point is that by virtue of the decorrelating properties of COPI we can very efficiently compute linear approximations of any sequence of nonlinear transformations in the network. This allows for direct visualization of features deep in the network (which is otherwise nontrivial) and direct compression of networks by replacing nonlinear transformation by linear approximations, with direct applications for tiny ML / efficient deep learning. I.e. after training a deep network once, we get the set of all shallower networks essentially for free by employing the linear approximation.
>
> Concerning point (4), the suggestions for figure modifications have all been integrated in the revised manuscript. We changed the scale of the y axes in the results figure and increased the size of the filters.

---

> > ### Comment · Reviewer_UPjV · 2023-01-12
> > **Final recommendation**
> >
> > I issued my final recommendation. Thank you for your response, I do not think that moving section to the main text is a good strategy to make the paper clearer. The paper was already rather confusing so I would have recommended instead to improve the text consistency (possibly by making it shorter).
> >
> > I still find the high level framing quite confusing (why is that parameter inference and not just another of learning rule?), and the motivation behind this algorithm is not clearly described (what is the criteria defining biological plausibility and why does it matters ?) but the derivations are correct and the simulations support that some ideas are valuable.
> >
> > I will remember in particular the idea of combining decorrelation and gradient descent in the weight update to simplify the learning rule.

---

### Review · Reviewer_ZrgT · 2022-12-05

**Summary Of Contributions:**

The authors propose a novel method for learning in neural networks. This method can learn directly and locally at each synapse from the activities (and desired activities) of the neurons, assuming that 1) desired activities (or desired directions of change in activities) are available for each neuron, and 2) inputs are uncorrelated. The method essentially performs Hebbian learning between inputs and desired outputs, with a weight decay term modulated by the variance of each input.

The authors propose a method to learn decorrelating layers that decorrelate the output of each main layer, facilitating assumption 2.

Various experiments suggest that the method is competitive with backpropagation with Adam.


**Audience:**

Yes

**Broader Impact Concerns:**

I do not see any broader impact concerns.

**Claims And Evidence:**

No

**Requested Changes:**

- Please explain what exactly COPI brings, when  it requires access to desired directions of changes in activities that already allow for local, Hebbian-like learning.

- In experiments, include a weight decay component in BP, comparable to that of COPI, and report results.

- Last paragraph on p.6: please clarify the statement that decorrelation improves performance also for BP, which seems to conflict with my understanding of Figure 1a and (in the long run) 1b.

- I don't understand the visualization part (second to last paragraph of p.  8). Why does decorrelation help here ? How
exactly is Figure 2b generated?

**Strengths And Weaknesses:**

Strength:
- The method is new, to my knowledge. The problem of plausible learning methods  is important.


Weaknesses:
- The method seems to make some requirement that put its purpose into question: if the required information described in the paper is provided, then traditional methods become local Hebbian-like learning rules, making the proposed method seemingly redundant?

- Secondarily, it is not clear that the experiments are fair, because IIUC they oppose a method COPI with weight decay against BP without weight decay.

In more detail:

Authors describe COPI as a local rule, that (by assuming decorrelation) can learn directly from local activities and desired/target activities at any node.

But the way these desired activities are obtained is by providing a desired *direction*, or *increment*, much like the error gradient
(over the weights) of BP, or the surrogate gradient of FB alignment. This desired direction of change is called little-delta in the text, in accordance with existing literature on gradient descent.

But if you have such a little-delta (desired direction of change in activity), then plain gradient learning is already purely local and
indeed "Hebbian", with the delta rather than the activities: deltaW = x * little-delta.

What, then, is gained at all by using the more complicated COPI ?

IIUC the specific form of COPI, and especially its assumption of decorrelated inputs, is only necessary because it tries to express
learning as a function of raw activities and desired activities themselves, rather than the desired delta. Indeed this seems to be
exactly what the "Correspondence to stochastic gradient descent" section says: the W(xx^T) term only pops up when we rewrite the plain
gradient update, dW = delta * x, as a function of actual activities a (and desired activities a + delta) instead.

But then the method assumes that we already have access to the deltas! So what have we gained by using the added complexity and assumptions of COPI?

Experiments are interesting, but difficult to interpret:

The authors claim the decorrelation layer improves performance across the board. But BP with Adam seems to outperform BP with decorrelation in terms of test error/accuracy in Figure 1a and (eventually) 1b? I may be missing something.

Authors observe that COPI beats BP (when both have similarly trained decorrelation layers) and attribute this to COPI having " the built-in
assumption that the inputs have a decorrelated form".

But IIUC COPI includes a weight decay term. Was there a weight decay in the BP version?  If there is no weight decay in the BP experiments, this might be a source  of the difference in results. This is particularly acute since at least some of
the failure of BP in the figures seems to be overfitting (test error increasing), and also considering the large size of the networks.

---

> ### Author Response · Authors · 2023-01-10
> **Clarifications and changes in response to this review**
>
> Thank you for your consideration, we attempt to give some clarifications, along with manuscript updates below.
>
> Concerning point (1), we would like to stress that there are important differences between COPI and BP.
>
> In terms of its derivation, we arrive at an expression for the optimal weight change without relying on gradient descent. That is, the derivation holds for any nudging of activity in the right direction. Equation (6) in the updated manuscript makes the distinction between BP and COPI clear. If we rewrite a single-layer BP update then we obtain an equation which looks like COPI with the important exception that the weight decay is right-multiplied by a full full rather than diagonal matrix. The diagonal matrix in COPI is obtained by applying the (BIO-)COPI rule to the decorrelating matrix R. Note that this online decorrelation process is a novel contribution and crucial to make the COPI algorithm work in practice.
>
> From an empirical perspective, COPI is shown to converge much more rapidly than BP (see Figure 1). That is, we demonstrate in the paper from first principles how to obtain much faster convergence rates compared to what is being used in practice (i.e., BP with adaptive optimizers). Moreover, the induced decorrelated representations lead to new approaches for feature visualisation and network compression.
>
> Of course, one could try to approximate this behaviour using BP. This is in fact done by combining BP with the COPI decorrelation rule. This is also shown to be effective yet it achieves lower performance and in of itself fundamentally relies on the theoretical machinery developed in this paper to understand why it is effective and how to make it work in practice (i.e. the use of the COPI rule to infer the decorrelating matrix).
>
> Regarding Hebbian updates, please note the following: if one were to rewrite error signals into a mixed system state (so that activities encode errors), then the update given by SGD (eq. 6) has a Hebbian component (pre and post activities multiplied). However, this requires a weight decay matrix multiplied by a full matrix which is the outer-product of the input vector with itself. This matrix is not locally available. In the special case of decorrelated inputs, this matrix becomes diagonal and one arrives at the COPI rule which does not require non-local information. To make this explicit, we added the following to Section 2.6:
>
> This update looks similar to the (stochastic) COPI update $\Delta_W^{copi} = (a + \alpha \delta^{sgd}) x^\top - W {diag}\left(x^2 \right )$. The key difference, however, is that, in contrast to SGD, COPI ensures that the inputs are decorrelated, as realized by the COPI update $\Delta_R^{copi}$. Therefore, the weight decay term is unaffected by sample-wise input cross-correlations. The weight decay term for COPI is Hebbian in nature since the update of $w_{ij}$  relies on the $[W {diag}\left(x^2\right)]{ij} = w_{ij} x_i x_j$, which is local to the synapse. In contrast, SGD is non-Hebbian in nature since it relies on $[W(x x^\top)]{ij} = \sum_k w_{ik} x_k x_j$, which is not local to the synapse.
>
> Concerning point (2), it is important to note that for COPI the weight decay appears as a result of the mathematical derivation of the algorithm and not as an additional penalty term. That is, COPI consists of a theoretically derived balancing of positive weight increase and weight decay. In this sense it does not play the traditional role of a weight decay as a means to prevent overfitting. In fact, adding such an explicit weight decay (e.g. by scaling the theoretically derived weight decay) would underestimate the weights when running the COPI algorithm. Therefore, it would not be appropriate to compare to BP with an explicit weight decay.
>
> Concerning point (3), we agree that the paragraph which is referred to was confusing as we collapsed across methods, training/testing, datasets and error signals. We rewrote that paragraph and now explicitly describe individual results. The competitiveness of COPI compared to BP remains although differences can be negligible. Note that the difference between conventional (adam-optimized) BP and COPI is very significant when comparing convergence speed as we mention later in that section.
>
> Concerning point (4), we also understood from Reviewer UPjV’s comment that the added benefit of COPI in terms of feature visualisation and network compression was hard to follow. We now describe these benefits earlier in the Introduction:
>
> Moreover, as will be shown, COPI allows for efficient linear approximations that facilitate feature visualisation and network compression. Hence, it also provides benefits in terms of interpretable and efficient deep learning.
>
> Furthermore, we moved the Appendix to a new Methods Section 2.7 and rewrote it to make it more intelligible. The corresponding Results section has been updated accordingly and more information has been added on how the linear transformations are being used.

---

> > ### Comment · Reviewer_ZrgT · 2023-01-10
> > **Thank you - addition needed**
> >
> > Thank you  for your response.
> >
> > From this comment and the one above, I believe my observation is  correct: COPI is only needed  if we prevent the weight-change mechanism from accessing  the "desired directions of change" $\delta_l$, which we use to compute the "target activities" $z_l = a_l + \alpha\delta_l$ (p. 4). That is, we  assume the plasticity system can only "see" perturbed activities  $a + \alpha\delta_l$, but not the $\delta_l$ themselves, even though these $\delta_l$ are necessary for  computing the perturbed activities!
> >
> > The only justification  I can see for this is aiming at some form of biological plausibility (though I note several works in computational  neuroscience propose to extract the guiding perturbation by temporal subtraction, see e.g. Hoerzer et al. Cerebral Cortex 2014, Legenstein et al. Journal of Neuroscience 2010, Miconi eLife 2017).
> >
> > This seems to be a somewhat artificial  restriction for most readers of a "machine learning" journal! However, I understand that value judgements about importance and impact should not prevent publication of an otherwise technically correct paper (which the present paper seems to be).
> >
> > Thus, I would not oppose publication, but I **require** that a paragraph similar to the following should be added in section 2.5:
> >
> > >  Note  that if  we can directly access the $\delta_l$, then the optimal  weight change to push future responses in the desired direction $\delta_l$ is simply $\delta_lx^\top$. However, here we assume that the weight-modification process cannot directly access $\delta_l$, but only "sees" the perturbed activities $z_l =a_l + \alpha\delta_l$. In this case, as explained above, COPI is necessary to produce correct weight changes and push future responses towards these target activities.
> >
> > The reason why I require such a paragraph is  that readers less familiar with this subject might not realize that COPI is not needed if $\delta_l$ are accessible. A paragraph similar to the above seems necessary to clearly  explain the significance  of the work to readers.
> >
> > Other than this, my other (more minor) concerns have been largely addressed.

---

> > > ### Author Response · Authors · 2023-01-23
> > > **Addition included**
> > >
> > > Thank you for your input. Indeed such a paragraph clarifies the specific use case for COPI and we appreciate your reasoning for including it. As proposed, we have added this exact paragrph in section 2.5 below the definition of the target states.

---

### Review · Reviewer_hXBa · 2022-12-28

**Summary Of Contributions:**

The paper proposes a novel learning algorithm to infer parameters of an artificial neural network. In this framework, learning can be established by updating the forward weight values based on observed local neuron activities, under the constraint that layer-wise neural inputs should be decorrelated via learned lateral weights (hence the name "constrained parameter inference": COPI). Proposed algorithm is also agnostic to the used credit assignment method to compute the layer-wise learning signal, where more bio-plausible local methods can be explored besides a standard gradient-based error signal. Experiments are performed with fully-connected networks on the MNIST and CIFAR-10 tasks, and COPI is compared against standard backpropagation based parameter optimization. Further contributions on using the outcome of the decorrelation constraint for network compression are discussed.


**Audience:**

Yes

**Broader Impact Concerns:**

No specific concern.


**Claims And Evidence:**

Yes

**Requested Changes:**

1) It is not really clear to what extent the introduction of target propagation (TP) is relevant in the manuscript. Since it is introduced as a compatible credit assignment approach here, can the authors add in simulations (Fig 1 & Table 1) and show how well it works when one simulates COPI together with TP-based perturbations?

2) Independent of the network compression experiments from Sec 3.2, can the authors demonstrate how important is the layer-wise depth of the FC network to learning with COPI from scratch? The networks in Fig 1, especially for MNIST, can be perhaps simulated with level-wise shallower architectures for this experiment.

3) Authors' proposal in Appendix D remains empirically untouched. How well does this approximation perform?

4) One of the interesting contributions is the network compression capability that arises from the produced linear approximations of receptive fields. However without addressing Appendix F, it is not possible to grasp how does this really work in Section 3.2. I believe a restructuring of the manuscript to address this would be helpful.

5) I found the algorithmic summary in Appendix A quite useful. I think moving Algorithm 1 to the main manuscript might be beneficial in depicting how subsections of Sec 2 come together in forming the learning algorithm that is mainly proposed in the manuscript.

**Strengths And Weaknesses:**

Strengths: The paper is clearly written, the derived learning algorithm is quite simple and yet effective. There are nice and clearly drawn bio-plausibility relationships throughout the manuscript regarding e.g., the local learning rules, or lateral weight updates to decorrelate layer-wise inputs.

Weaknesses: I believe the major weakness that requires the most attention is the coverage of experimental simulations. Currently, not all aspects of the theoretically discussed and presented components of the learning algorithm are evaluated in the manuscript.

---

> ### Author Response · Authors · 2023-01-10
> **Summary of changes in response to review**
>
> Thank you, reviewer, for your attention and comments.
>
> Concerning point (1), TP is not essential for the manuscript as it is just an example of another error signal which can be plugged in. To improve the flow of the paper, we decided to simplify the Error Signals section and place less emphasis on TP and other variants. We also removed the distinction between delta^{sgd} and delta^{bp} since it was unnecessary and confusing.
>
> Concerning point (2), we added the requested results for different network depths to the Appendix. Note further that our aim is not to identify the optimal network depth but rather to show effective credit assignment in multi-layer neural networks. In Section 3.1 we now state:
>
> Note further that results generalize to different network depths, as shown in Appendix C, as well as to other loss functions, as shown in Appendix D for the cross-entropy loss.
>
> Concerning point (3), we now integrated the description of the more biologically plausible variant of COPI in Section 2.4 and added new simulation results using BIO-COPI in Section 3. Performance is close to identical to that of regular COPI. We also updated Section 2.5 to refer to the weight-transport problem and Section 2.6 to compare Hebbian and non-Hebbian updates of COPI and SGD, respectively.
>
> Concerning point (4), we now integrated Appendix F in the main text Section 2.7. This strongly improved the flow of the paper. We also updated Section 3.2 accordingly which now makes it easier to follow.
>
> Concerning point (5), the algorithmic summary was moved to the main text and updated (Section 2.6).

---

> > ### Comment · Reviewer_hXBa · 2023-01-17
> > **thanks for the responses**
> >
> > Thanks to the authors for their responses and comments.
> >
> > I find the performed revisions sufficient to my understanding of the proposed algorithm here. Overall, derivations in the paper appear to be technically correct and the experiments convincingly support the paper's reasoning. I submitted my final recommendation accordingly.

---

### Comment · Reviewer_ZrgT · 2022-12-29
**Need some explanation**

I would appreciate help from reviewers/authors/editors in understanding why the proposed method is not redundant?

1- IIUC it assumes that we have access to desired directions of changes for the response of each neuron (little-delta in section 2.5)

2- But if we have that, then the *exact* gradient update for w is simply the Hebbian-like product, delta-w =  x * little-delta !  Which is entirely local and biologically plausible.

3- So why do we need COPI and its complicated machinery?

As mentioned in my review, it seems to me (maybe I'm wrong) that the reason why the authors came up with the COPI method in the first place is that they started writing their equations by assuming availability of desired *activities* (rather than desired *directions of change in activities*). In that case, if you don't want to mess with subtractions, then the correct update is indeed more complicated and involves correcting for correlations, as shown by the authors' derivation.  But in practice, they acknowledge that desired directions of change in activities are more easily available than desired activities.

This is my understanding of the paper. Of course, I may be entirely off. If so I would appreciate a simple, few-sentences explanation of what I'm missing.

---

> ### Author Response · Authors · 2023-01-10
> **Clarification: our updates assume an activity space for updates.**
>
> Dear reviewer ZrgT, we attempted to address these points in the our response to your review (below) but also attempt to be more explicit below.
>
> In our paper, we assume a system in which the backward and forward passes co-exist in an activity space. This truly makes things 'Hebbian' since updates would then be input and output activity based. However, in point 2, your computation for delta includes an input which is the forward pass but an output which is the backward pass (not activity based or co-existent anymore). Our desire for the co-existence of backward and forward passes is that we designed our learning rule upon the principle that parameters might infer their own values based upon an observed 'desired' output. Thus, COPI is also agnostic to the perturbation type (it's not limited for use with BP gradient directions) and simply describes the required setup for parameters to locally infer their own value for a desired output activity shift. Given this, acting in activity space is a design feature of our algorithm.
>
> More specifically for the case when combining with BP:  we describe a case in which our forward and backward passes are mixed, and little-delta is $\frac{d loss}{ d a}$ is not solely available. The Hebbian term between input and (output + little-delta) then requires an additional non-local weight decay to match with sgd. In our COPI work, we derive an inference method that directly informs us to diagonalize the autocorrelation matrix allowing for an update based upon local operations instead. Our derivation also includes a mechanism to learn the decorrelated inputs, which is the COPI rule applied to R which is purely Hebbian. This is why we need the COPI machinery which is developed in this paper, though it has a number of other benefits. Also, note that this machinery is not necessarily much more complicated than BP since it only requires the existing perturbation signal and a decorrelating mechanism.

---

> > ### Comment · Reviewer_ZrgT · 2023-01-10
> > **Thank you**
> >
> > Thank you for your response. See my response below: https://openreview.net/forum?id=CUDdbTT1QC&noteId=TWTCqoH0HD

---

### Decision · Action_Editors · 2023-01-23

**Recommendation:** Accept as is

**Comment:**

All reviewers agreed that the revised version of the manuscript can be accepted for publication.
There was some discussion on restrictive assumptions of the work. It was noted that the proposed algorithm is useful if the plasticity rule does not have access of the plasticity process to desired directions of activity changes and a paragraph which clarifies this was added in a final revision as demanded. As this was clarified, I recommend acceptance.

**Audience:**

The manuscript will be of interest for readers interested in efficient biologically-plausible learning algorithms for neural networks.

**Claims And Evidence:**

The authors propose a novel learning algorithm for neural networks. Derivations for the update rules are provided. The reviewers note that these derivations are correct. Experiments on MNIST and CIFAR-10 show the effectiveness of the method.